crystallography

metastable zone width, solubility, nucleation kinetics, borax

**Author for correspondence:**
Yaping Dong
e-mail: dongyaping@hotmail.com

This article has been edited by the Royal Society of Chemistry, including the commissioning, peer review process and editorial aspects up to the point of acceptance.

# Effects of $CO_3^{2-}$ and $OH^-$ on the solubility, metastable zone width and nucleation kinetics of borax decahydrate

Jing Chen[1,2,3], Jiaoyu Peng[1,2], Xingpeng Wang[1,2,3], Yaping Dong[1,2] and Wu Li[1,2]

[1]Key Laboratory of Comprehensive and Highly Efficient Utilization of Salt Lake Resources, Qinghai Institute of Salt Lakes, Chinese Academy of Sciences, 810008 Xining, Qinghai, People's Republic of China
[2]Engineering and Technology Research Center of Comprehensive Utilization of Salt Lake Resources, 810008 Xining, People's Republic of China
[3]University of Chinese Academy of Sciences, 100039 Beijing, People's Republic of China

YD, 0000-0003-1903-4529

Measurements of the solubility and metastable zone width (MZW) of borax decahydrate in sodium carbonate and sodium hydroxide aqueous were obtained. The onsets of nucleation were detected by the turbidity technique with the temperature range from 285 to 315 K. The results showed that the solubility of borax gradually decreased and the MZW broadened with the mass percentage of sodium carbonate increasing from 0% up to 9.22%. Correspondingly, the solubility and MZW had the same trend with the addition of sodium hydroxide. Meanwhile, the nucleation parameters of borax were determined and analysed to explain the trends obtained. Applying the classical three-dimensional nucleation theory approach, it was found that the addition of carbonate and hydroxide ions led to the values of solid–liquid interfacial energy ($\gamma$) increasing, which indicated the $CO_3^{2-}$ and $OH^-$ ions adsorbed on the nuclei but suppressed nucleation rate.

## 1. Introduction

Boron compounds have unique advantages in their porosity, density and thermal stability, leading to potential applications for hydrogen storage, filtration, catalysis and optoelectronics [1–3]. In geological formations, boron compounds have different kinds of existing forms, such as sassoline, borax, ulexite and colemanite [1]. There is a growing interest in the crystallization of liquid and solid boron due to their long-term exploitation [4].

Brine is a major source of boron supply, and borax decahydrate ($Na_2B_4O_7 \cdot 10H_2O$) is a typical crystalline product from brine [5]. For the investigation of crystallization of borax from brine, it is extremely important to know its solubility and MZW as functions of temperature and presence of other salts in the solution [6].

The major variables affecting the crystallization of borax from its solutions have been comprehensively investigated [7–9]. However, to obtain borax products from brine, evaporation processes are followed, which mainly depend on the species and quantity of coexisting compounds present in the solution [10]. Gurbuz & Ozdemir [11] investigated the effects of $Ca^{2+}$ and $Mg^{2+}$ by ultrasonic velocity technique. They found that trace amount of $Ca^{2+}$ and $Mg^{2+}$ had a slight effect on MZW of borax, but increasing the concentration will result in a reasonable increase. Peng [12–14] have investigated the influence of KCl, LiCl and $K_2SO_4$ on solubility and MZW using a laser technique. The results showed that KCl, LiCl and $K_2SO_4$ had a salt-in effect on borax. The MZW broadened with increasing the LiCl and $K_2SO_4$, but decreasing by KCl. The opposite effects of $K_2SO_4$ and KCl were attributed to the different mechanisms of $Cl^-$ and $SO_4^{2-}$, suggesting that the effects of anions cannot be neglected.

Generally, there are several anions in brine, such as $Cl^-$, $SO_4^{2-}$, $CO_3^{2-}$ and $OH^-$ [15]. $CO_3^{2-}$ ion is one of the main components of carbonate-type brine which always contains a large amount of boron. A typical example is Zabuye salt lake [16]. Different concentrations of $CO_3^{2-}$ lead to different pH values, thus affecting the solubility and MZW of borax in brine. Although there are several reports on the solubility and MZW of borax, the results are still inadequate and roughly compared with cations. Therefore, the aim of this study was to investigate the influence of $CO_3^{2-}$ and pH on solubility and the MZW of borax using the polythermal method. Then the experimental MZW data of different cooling/heating rates $R$ were analysed and discussed by three-dimensional nucleation theory.

# 2. Experiment section

## 2.1. Material and methods

All of the chemical reagents used in this study are listed in table 1. $Na_2B_4O_7 \cdot 10H_2O$ was recrystallized with a purity more than 99.99%. Water (resistivity, $18.25\ M\Omega\ cm^{-1}$) was deionized from a water purification system (UPT-II-20T, Chengdu Ultrapure Technology Co., Ltd) before experiments.

The experimental set-up is shown in figure 1. A turbidity meter was employed to detect nucleation/dissolution. The temperature of prepared solution was measured using the digital thermometer with precision of $\pm 0.1°C$. Cooling rates control was accomplished using a Crystal SCAN with four parallel reactors (E1061, HEL, UK) containing systems for temperature control and computer processing as well as a crystallizer assisted with programmable thermostatic bath (FP50-ME, Julabo, Germany). The crystallizer was a 100 ml glass vessel with an internal overhead stirrer, temperature sensor and turbidity sensor. Besides, the crystallizer was made air-tight so that the loss of solvent due to evaporation could be minimized. The X-ray diffraction (XRD) analysis (X'Pert PRO, 2006 PANalytical) was used to confirm the identity of the solid phase crystallizing from the solutions. The pH was measured by pH meter (S470 Seven Excellence, Mettler Toledo) with precision of $\pm 0.05$.

## 2.2. Solubility and MZW determination

The determination of the solubility and MZW of the borax in sodium carbonate solutions was carried out with a temperature range from 285 to 315 K according to the conventional polythermal method. Firstly, 60 g of mixture was placed into a 100 ml crystallizer. Then, the mixture was heated with a given rate above the saturation temperature for 10 min to ensure complete dissolution of the solid phase. Finally, the solution was cooled down with the same constant rate until the first visible nucleus appeared, which can be detected by a sudden increase in turbidity. The corresponding temperature at the point of nucleation and dissolution was recorded as $T_1$ and $T_2$, respectively. The above steps were repeated at five cooling/heating rates of 55, 45, 35, 25 and 15 K h$^{-1}$ and at constant impeller speed of 300 r.p.m. The concentration of borax and sodium carbonate in the solution was obtained by titration. The faster the heating rates were, the higher the measured dissolution temperature was. Therefore, the saturation temperature $T_0$ of borax can be obtained by extrapolating the $T_2$–$R$ curve to a virtual heating rate of 'zero'. The metastable zone width (MZW) of borax is represented by the maximum undercooling $\Delta T_{max}$ ($\Delta T_{max} = T_0 - T_1$).

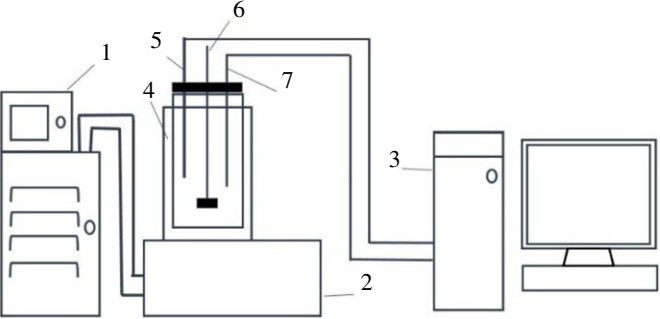

**Figure 1.** Schematic representation of experimental set-up. (1, low constant temperature bath; 2, temperature control system; 3, a computer processing system; 4, crystallizer; 5, turbidity sensor; 6, overhead stirring; 7, temperature sensor.)

**Table 1.** Chemical reagents employed in the experiment.

| chemical name | formula | provider | purity |
|---|---|---|---|
| borax decahydrate | $Na_2B_4O_7 \cdot 10H_2O$ | Tianjin Damao Chemical Reagent Factory | $\geq$99.99% |
| sodium carbonate anhydrous | $Na_2CO_3$ | Tianjin Yongda Chemical Reagent Development Center | $\geq$99.95% |
| sodium hydroxide | NaOH | Tianjin Kermel Chemical Reagent Development Center | $\geq$99.95% |

## 2.3. Chemical analysis

The boron content was determined by mannitol conversion acid$-$base titration. The $CO_3^{2-}$ ion concentration was determined by adding 0.05 mol l$^{-1}$ HCl, and using methyl red-bromcresol green as indicator. The accuracy of these analyses was about 0.1%. All of the estimated uncertainties of the research are listed in table 2.

# 3. Results and discussion

## 3.1. XRD analysis

The XRD patterns of borax obtained from pure water, sodium hydroxide and sodium carbonate were investigated, respectively, shown in figure 2. It manifests the XRD patterns are identical and well indexed to borax without any impurities, according to the reference data JCPDS 75-1078.

## 3.2. Solubility

The solubility of borax in different mass percentages of sodium carbonate (0.0–9.22%) in aqueous solution was determined. The obtained experimental solubility data are demonstrated graphically in figure 3. It can be observed in figure 3$a$ that the solubility of borax increased with temperature, which can be attributed to the thermal motion of molecule. Besides, the addition of sodium carbonate leads to the solubility decrease. It could be the common ion effect that the addition of sodium carbonate releases $Na^+$ into the solution, making dissolution$-$precipitate equilibrium move towards the precipitate. Therefore, the solubility of borax decreases. It also can be seen that the solubility curves are roughly paralleled to each other, indicating that the increment of sodium carbonate causes a gradual decrement of borax solubility.

Furthermore, it can be seen that pH increases with the addition of $Na_2CO_3$ (from 0 to 9.22%), based on figure 3$a$. Therefore, pH range from 9.9 to 10.5 was selected to investigate by the addition of $Na_2CO_3$ and NaOH, respectively. As shown in figure 3$a$, the solubility of borax at $Na_2CO_3$ concentration of 9.22% (pH = 10.5) was lower than pH = 10.5 adjusted by NaOH, seen in figure 3$b$. It suggests that $CO_3^{2-}$ has more prominent effects on the decrement of borax solubility than $OH^-$.

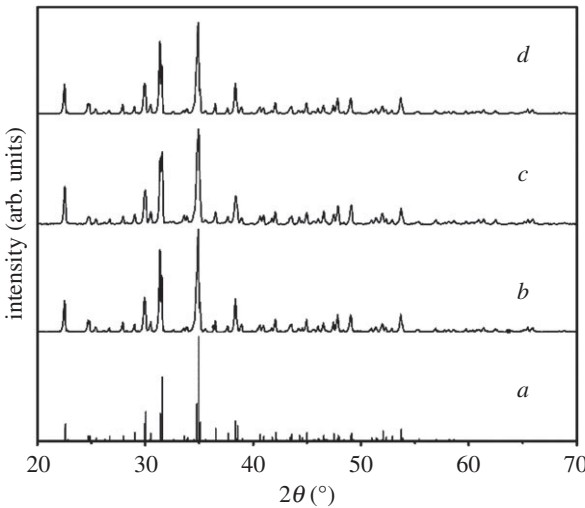

**Figure 2.** The XRD patterns of crystallized borax decahydrate: (*a*) standard; (*b*) crystallized in pure water; (*c*) crystallized in sodium hydroxide solution; (*d*) crystallized in sodium carbonate solution.

**Table 2.** Uncertainties of measurements estimated for this research.

| property | estimated uncertainty |
|---|---|
| solubility | $\pm 0.05$ g of 100 g of $H_2O$ |
| temperature | $\pm 0.06°C$ |
| pH | $\pm 0.03$ |
| *w* % | $\pm 0.08$ |

## 3.3. Thermodynamic properties of borax

Dissolution enthalpy, $\Delta_{dis}H$, and dissolution entropy, $\Delta_{dis}S$, are important to investigate the dissolution behaviour of the solute in different solvents. When the solubility of borax in sodium carbonate and sodium hydroxide solution at different temperatures is available, then the values of $\Delta_{dis}H$ and $\Delta_{dis}S$ can be determined from the van't Hoff equation as follows:

$$\ln x = \frac{-\Delta_{dis}H}{R_G T_0} + \frac{\Delta_{dis}S}{R_G},\tag{3.1}$$

Where $\Delta_{dis}H$ and $\Delta_{dis}S$ are the dissolution enthalpy and entropy, respectively, $R_G$ is gas constant ($8.314 \, J \, mol^{-1} \, K^{-1}$) and $x$ is the mole fraction of borax. The van't Hoff plots shown in figure 4 are obtained from the linear fit of ln $x$ versus $1/T_0$. Then the dissolution enthalpy and entropy of borax which are shown in table 3 can be calculated from the slope and the interception of these plots.

The values of dissolution enthalpy and entropy of borax are 34.11 kJ mol$^{-1}$ and 61.14 J mol$^{-1}$ K$^{-1}$ [17] in the literature, which are in good agreement with our experimental data. It can be found from table 3 that the dissolution enthalpy and entropy are positive, which indicates that the dissolution is always endothermic and entropy driven. It also can be found that the mixture with more sodium carbonate and sodium hydroxide has lower solubility but higher values of $\Delta_{dis}H$ and $\Delta_{dis}S$, which is consistent with general thermodynamic principles [18].

## 3.4. Metastable zone width

The MZW data of borax against saturation temperature at different mass percentages of sodium carbonate are given in figure 5a. It is clear that the MZW becomes broad with the increase of sodium carbonate. The effects are more significant at higher mass percentages, but have little effect with mass percentages of sodium carbonate below 5.31%. The effects could have two possible explanations. One is that carbonate ions with large size may block the active growth sites of the

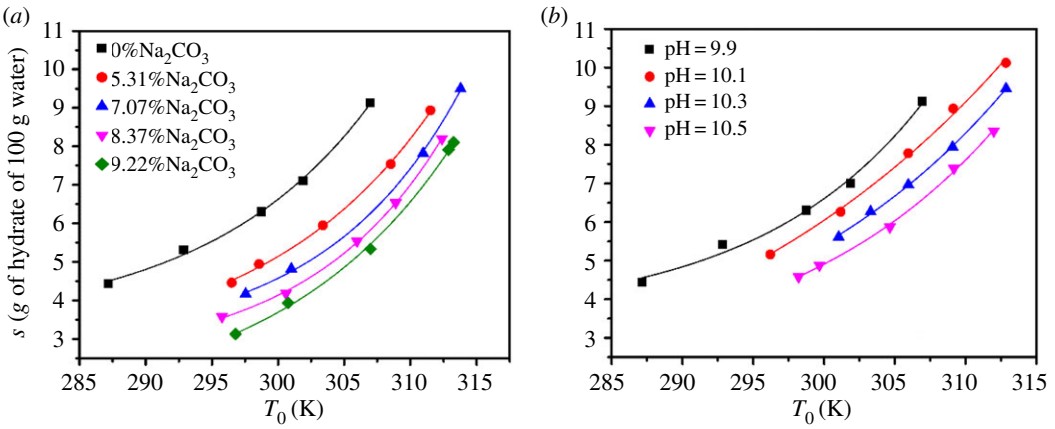

**Figure 3.** Solubility of borax (*a*) effects by $CO_3^{2-}$; (*b*) effects of pH adjusted by NaOH.

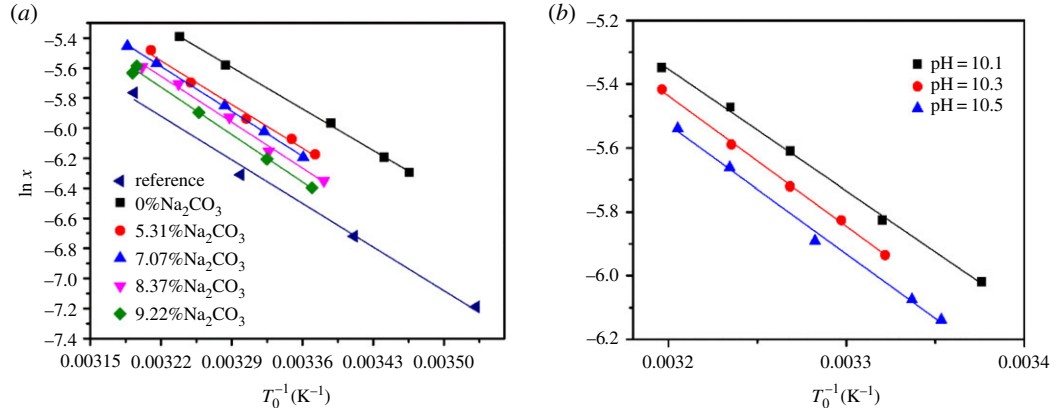

**Figure 4.** The van't Hoff plots of ln *x* versus $1/T_0$. (*a*) In sodium carbonate solution; (*b*) in sodium hydroxide solution.

**Table 3.** Dissolution enthalpy and entropy of borax in sodium carbonate solutions.

| factor | value | $\Delta_{dis}H$ (kJ mol$^{-1}$) | $\Delta_{dis}S$ (J mol$^{-1}$ K$^{-1}$) | $R^2$ |
|---|---|---|---|---|
| *w* % (Na$_2$CO$_3$) | 0.00 [17] | 34.11 | 61.14 | 0.9909 |
| | 0.00 | 32.84 | 61.51 | 0.9914 |
| | 5.31 | 34.12 | 64.21 | 0.9908 |
| | 7.07 | 36.66 | 71.80 | 0.9987 |
| | 8.37 | 37.37 | 73.27 | 0.9694 |
| | 9.22 | 37.99 | 74.38 | 0.9954 |
| pH | 10.1 | 31.72 | 57.01 | 0.9977 |
| | 10.3 | 33.62 | 61.62 | 0.9985 |
| | 10.5 | 33.92 | 62.33 | 0.9942 |

nuclei forming in bulk solution due to steric effect, which might depend on concentrations. When presented in relatively small concentrations, these ions suppress nucleation slightly. Therefore, it might be seen that the higher mass percentages of carbonate ions, the more powerful the inhibiting effects are. Another possible reason is that carbonate ions might act as surface active agents, rendering the nuclei inactive [19].

The results of the influences of pH on MZW are also given in figure 5*b*. It should be noted that the MZW of borax is larger at higher pH. This could be explained by the relationship between the pH and polyborate speciation [2]. Based on the reports [20,21], the speciation of boron strongly depends on

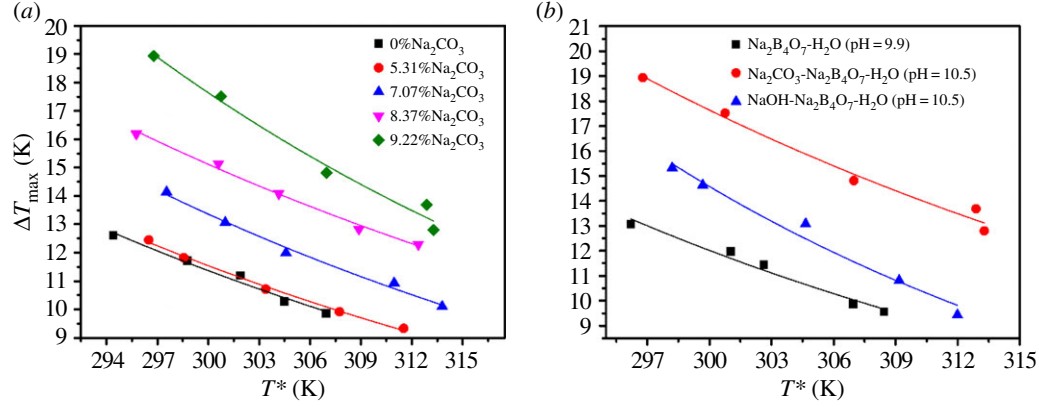

**Figure 5.** Changes in MZW ($R = 55$ K/h): ($a$) at different mass percentages of sodium carbonate solutions; ($b$) at same pH.

**Figure 6.** The plot of $(T_0/\Delta T_{max})^2$ versus $\ln R$ for borax in different mass percentages of sodium carbonate: ($a$) 0.00% $Na_2CO_3$; ($b$) 5.31% $Na_2CO_3$; ($c$) 7.09% $Na_2CO_3$ ; ($d$) 8.38% $Na_2CO_3$; ($e$) 9.22% $Na_2CO_3$.

the chemical medium, especially the pH. It appears that tetraborate species exist in solution for pH ranging from 7 to 12 and are abundant at about pH = 10. With the higher pH, the concentration of $B_4O_5(OH)_4^{2-}$ will decrease, which makes nucleation more difficult. In consequence, the MZW is

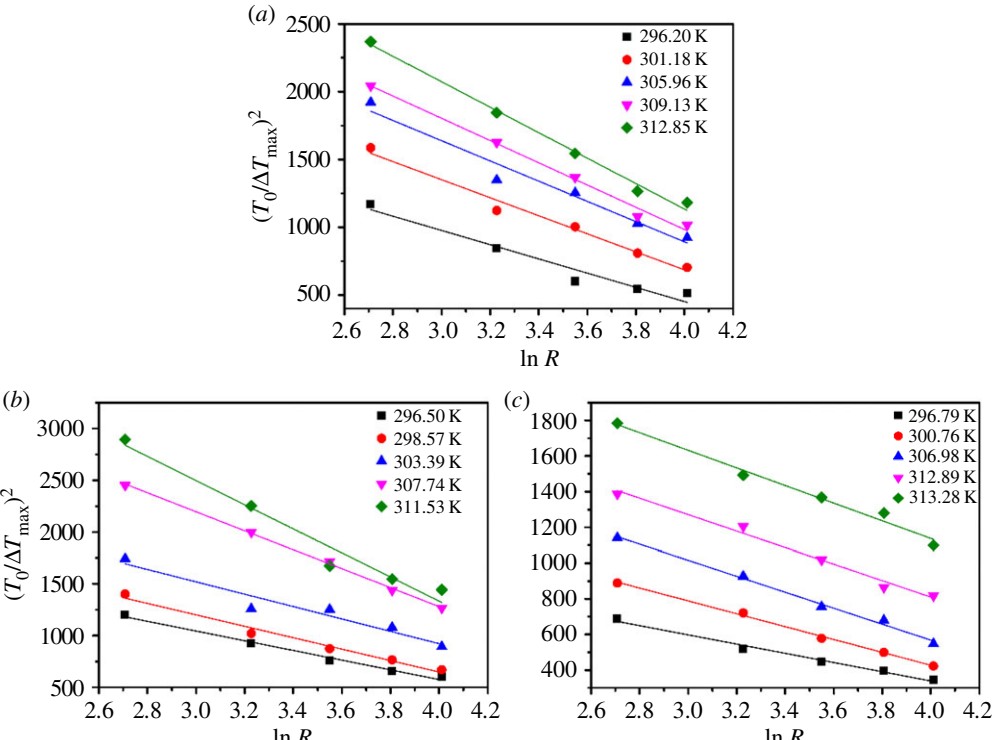

**Figure 7.** The plot of $(T_0/\Delta T_{max})^2$ versus $\ln R$ for borax at: (a) pH = 10.1; (b) pH = 10.3 ; (c) pH = 10.5

broadened. Besides, when the solution pH remains at 10.5, seen in figure 5b, it can be easily found that the MZW of borax in the $Na_2CO_3$–$NaB_4O_7$–$H_2O$ system is wider than that of the $NaOH$–$NaB_4O_7$–$H_2O$ system. It may be the impurities in the former system are $CO_3^{2-}$ and $OH^-$. Nevertheless, there is only $OH^-$ ion in the $NaOH$–$NaB_4O_7$–$H_2O$ system. The extra $CO_3^{2-}$ ion in sodium carbonate solution would retard nucleation so that the MZW is broadened.

## 3.5. Classical three-dimensional nucleation theory approach

The solid–liquid interfacial energy $\gamma$ is an important thermodynamic parameter that indicates the ability of the solute to crystallize from solution [22]. According to the classical three-dimensional nucleation theory, the relationship between MZW and cooling/heating rate $R$ can be represented as equation (3.2) [23,24]. Where $k_B$ is the Boltzmann's constant equal to $R_G/N_A$ ($N_A$ is the Avogadro number); where $A$ is a kinetic constant associated with the media of nucleation; $f$ is a constant expressing the number of nuclei in certain volume; $\Omega$ is the molecular volume, calculated by the density. Figures 6 and 7 present plots of $(T_0/\Delta T_{max})^2$ against $\ln R$ for borax in different percentage fractions of sodium carbonate and sodium hydroxide solutions according to equation (3.2). The values of $\gamma$ and $A$ can be calculated from the slope and the intercept by equations (3.3)–(3.5), respectively

$$\left(\frac{\Delta T_{max}}{T_0}\right)^2 = F - F_1 \ln R = F(1 - Z\ln R), \tag{3.2}$$

where $F$, $Z$ and $B$ represent slope, intercept and nucleation parameter, values are calculated by equations

$$F = \frac{1}{ZB}\left(\frac{\Delta_{dis}H}{R_G T_{lim}}\right)^2, \tag{3.3}$$

$$Z = \frac{F_1}{F} = \ln\left(\frac{f}{AT_0}\frac{\Delta H_s}{R_G T_{lim}}\right) \tag{3.4}$$

and

$$B = \frac{16\pi}{3}\left(\frac{\gamma\Omega^{1/3}}{k_B T_{lim}}\right)^3. \tag{3.5}$$

As shown in table 4, the estimated solid–liquid interfacial energy $\gamma$ in pure water is 1.81 mJ$^{-3}$ m$^{-2}$, which agrees well with the data 1.7 in the previously published literature [25]. It should be noted that the

**Table 4.** Values of kinetic parameters of borax at different mass percentages of sodium carbonate estimated using classical three-dimensional nucleation theory (units: $T_0$, K; $\gamma$, mJ$^{-3}$ m$^{-2}$; $A$, $10^{24}$m$^{-3}$ h$^{-1}$).

| w% (Na$_2$CO$_3$) | $T_0$ | nucleation equation | $A$ | $\gamma$ | $R^2$ |
|---|---|---|---|---|---|
| 0.00 | 294.41 | $(T_0/\Delta T_{max})^2 = -362.37 \ln R + 1849.18$ | 2.80 | 2.52 | 0.9937 |
| | 298.77 | $(T_0/\Delta T_{max})^2 = -472.07 \ln R + 2364.52$ | 2.90 | 2.31 | 0.9994 |
| | 301.89 | $(T_0/\Delta T_{max})^2 = -547.25 \ln R + 2800.16$ | 3.38 | 2.22 | 0.9933 |
| | 304.48 | $(T_0/\Delta T_{max})^2 = -716.02 \ln R + 3557.29$ | 3.89 | 2.04 | 0.9953 |
| | 306.95 | $(T_0/\Delta T_{max})^2 = -1041.34 \ln R + 5125.39$ | 4.64 | 1.81 | 0.9990 |
| 5.31 | 296.50 | $(T_0/\Delta T_{max})^2 = -140.56 \ln R + 1131.58$ | 3.03 | 3.45 | 0.9966 |
| | 298.57 | $(T_0/\Delta T_{max})^2 = -165.62 \ln R + 1300.76$ | 3.14 | 3.27 | 0.9996 |
| | 303.39 | $(T_0/\Delta T_{max})^2 = -220.39 \ln R + 1688.51$ | 3.63 | 2.99 | 0.9963 |
| | 307.74 | $(T_0/\Delta T_{max})^2 = -308.56 \ln R + 202.60$ | 4.16 | 2.69 | 0.9982 |
| | 311.53 | $(T_0/\Delta T_{max})^2 = -354.87 \ln R + 2546.63$ | 4.97 | 2.55 | 0.9959 |
| 7.09 | 297.55 | $(T_0/\Delta T_{max})^2 = -110.68 \ln R + 880.04$ | 2.79 | 3.80 | 0.9879 |
| | 301.00 | $(T_0/\Delta T_{max})^2 = -174.97 \ln R + 1232.95$ | 3.32 | 3.28 | 0.9989 |
| | 304.59 | $(T_0/\Delta T_{max})^2 = -207.83 \ln R + 1472.73$ | 3.94 | 3.12 | 0.9921 |
| | 310.97 | $(T_0/\Delta T_{max})^2 = -293.85 \ln R + 1971.71$ | 4.86 | 2.80 | 0.9901 |
| | 313.53 | $(T_0/\Delta T_{max})^2 = -352.01 \ln R + 2400.10$ | 5.28 | 2.64 | 0.9993 |
| 8.38 | 295.76 | $(T_0/\Delta T_{max})^2 = -92.97 \ln R + 701.67$ | 2.51 | 4.08 | 0.9917 |
| | 300.60 | $(T_0/\Delta T_{max})^2 = -125.94 \ln R + 899.25$ | 3.01 | 3.72 | 0.9940 |
| | 304.16 | $(T_0/\Delta T_{max})^2 = -134.00 \ln R + 1000.39$ | 3.67 | 3.66 | 0.9858 |
| | 308.89 | $(T_0/\Delta T_{max})^2 = -235.34 \ln R + 1521.09$ | 4.34 | 3.05 | 0.9936 |
| | 312.38 | $(T_0/\Delta T_{max})^2 = -259.56 \ln R + 1685.47$ | 4.74 | 2.97 | 0.9982 |
| 9.22 | 296.79 | $(T_0/\Delta T_{max})^2 = -84.17 \ln R + 579.43$ | 3.40 | 4.31 | 0.9902 |
| | 300.76 | $(T_0/\Delta T_{max})^2 = -104.23 \ln R + 706.29$ | 3.79 | 4.04 | 0.9881 |
| | 306.98 | $(T_0/\Delta T_{max})^2 = -159.14 \ln R + 1061.86$ | 4.04 | 3.54 | 0.9948 |
| | 312.89 | $(T_0/\Delta T_{max})^2 = -191.60 \ln R + 1291.13$ | 5.25 | 3.35 | 0.9988 |
| | 313.28 | $(T_0/\Delta T_{max})^2 = -212.14 \ln R + 1448.67$ | 5.40 | 3.24 | 0.9995 |

values of solid–liquid interfacial energy $\gamma$ decrease with an increase in saturation temperature $T_0$, but increase with the addition of Na$_2$CO$_3$, seen from table 4. Based on the reported articles, it is known that the increase in the value $\gamma$ will suppress nucleation rate and broaden the MZW. Basically, the higher interfacial energy means the bigger nuclear barrier which leads to the harder nucleation process. According to the data from table 4, it could be concluded that the adsorption of CO$_3^{2-}$ on the nucleus surface leads to the increase in $\gamma$ [26]. Furthermore, the solid–liquid interfacial energy $\gamma$ was less in NaOH–NaB$_4$O$_7$–H$_2$O system than that of Na$_2$CO$_3$–NaB$_4$O$_7$–H$_2$O system, seen from table 5. It can be explained by the fact that the charge of OH$^-$ is less than CO$_3^{2-}$ so that the adsorption is weaker.

# 4. Conclusion

The effects of sodium carbonate and sodium hydroxide on the solubility and MZW of borax have been studied at temperature ranging from 285 to 315 K using turbidity technique. A salting-out effect was observed under the larger mass percentages of the sodium carbonate and sodium hydroxide, which resulted in the lower solubility of borax. It was found that the addition of sodium carbonate broadened the MZW significantly, and the influence depended on concentration. It was believed that the addition of sodium carbonate adsorbed on nuclei and suppressed the activities of nuclei in the solution which enabled the larger MZW. In addition, the pH had an effect on polyborate species. The

**Table 5.** Values of kinetic parameters of borax at different pH values estimated using classical three-dimensional nucleation theory (units: $T_0$, K; $\gamma$, mJ$^{-3}$ m$^{-2}$; $A$, $10^{24}$ m$^{-3}$ h$^{-1}$).

| pH | $T_0$ | nucleation equation | $A$ | $\gamma$ | $R^2$ |
|---|---|---|---|---|---|
| 10.1 | 296.20 | $(T_0/\Delta T_{max})^2 = -534.51 \ln R + 2550.10$ | 10.47 | 2.16 | 0.9386 |
| | 301.18 | $(T_0/\Delta T_{max})^2 = -665.23 \ln R + 3346.68$ | 10.13 | 2.01 | 0.9760 |
| | 305.96 | $(T_0/\Delta T_{max})^2 = -745.82 \ln R + 3875.80$ | 9.84 | 1.95 | 0.9540 |
| | 309.13 | $(T_0/\Delta T_{max})^2 = -820.24 \ln R + 4264.34$ | 9.63 | 1.89 | 0.9884 |
| | 312.85 | $(T_0/\Delta T_{max})^2 = -938.72 \ln R + 4889.24$ | 9.39 | 1.82 | 0.9905 |
| 10.3 | 296.50 | $(T_0/\Delta T_{max})^2 = -468.49 \ln R + 2449.70$ | 10.25 | 2.26 | 0.9862 |
| | 298.57 | $(T_0/\Delta T_{max})^2 = -551.84 \ln R + 2855.87$ | 10.01 | 2.14 | 0.9754 |
| | 303.39 | $(T_0/\Delta T_{max})^2 = -597.27 \ln R + 3311.14$ | 9.97 | 2.10 | 0.9188 |
| | 307.74 | $(T_0/\Delta T_{max})^2 = -917.03 \ln R + 4947.78$ | 9.68 | 1.82 | 0.9983 |
| | 311.53 | $(T_0/\Delta T_{max})^2 = -1164.65 \ln R + 5991.67$ | 9.35 | 1.69 | 0.9525 |
| 10.5 | 296.79 | $(T_0/\Delta T_{max})^2 = -234.64 \ln R + 1280.51$ | 10.60 | 2.73 | 0.9850 |
| | 300.76 | $(T_0/\Delta T_{max})^2 = -361.28 \ln R + 1872.36$ | 10.39 | 2.45 | 0.9964 |
| | 306.98 | $(T_0/\Delta T_{max})^2 = -448.20 \ln R + 2360.97$ | 10.03 | 2.30 | 0.9918 |
| | 312.89 | $(T_0/\Delta T_{max})^2 = -461.23 \ln R + 2655.04$ | 9.85 | 2.29 | 0.9814 |
| | 313.28 | $(T_0/\Delta T_{max})^2 = -472.77 \ln R + 3068.73$ | 9.82 | 2.28 | 0.9140 |

increasing of pH could make the concentration of the tetraborate decrease, which led to the broadening of MZW. Finally, the obtained MZW data were analysed with the classical three-dimensional nucleation theory approach. The value of solute–solvent interfacial energy $\gamma$ increased as the mass percentage of sodium carbonate was larger. The investigation of these parameters is very useful for the design and development of a crystallization process.

Data accessibility. This article does not contain any additional data.

Authors' contributions. J.C. participated in all procedures including the design of the study, carrying out the laboratory work, data analysis and drafting the manuscript. J.P. participated in the design of the study and drafted the manuscript. X.W. carried out the analyses. Y.D. and W.L. conceived of the study. All authors gave final approval for publication.

Competing interests. The authors have no competing interests.

Funding. This work was supported by the National Key R&D Program of China (no. 2017YFC0602805); National Natural Science Foundation of Qaidam Salt lake Chemical Science Research Joint Fund (no. U1607103); National Natural Science Foundation for the Youth (no. 21501187); Enterprise Project (no. 2015-6300-101).

Acknowledgements. We are grateful to Dandan Gao and Shaoju Bian, who provided suggestions and encouragement during the research.

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
