## [Reviewer comments · Royal Society Open Science]

Review History

RSOS-181862.R0 (Original submission)

Review form: Reviewer 1

Is the manuscript scientifically sound in its present form?

No

Are the interpretations and conclusions justified by the results?

Yes

Is the language acceptable?

No

Is it clear how to access all supporting data?

Yes

Do you have any ethical concerns with this paper?

No

Have you any concerns about statistical analyses in this paper?

I do not feel qualified to assess the statistics

Recommendation?

Major revision is needed (please make suggestions in comments)

Comments to the Author(s)

The manuscript: Effects of CO₃²⁻ and OH⁻ on the Solubility, Metastable Zone Width, and Nucleation Kinetics of Borax Decahydrate written by Jing Chen, Jiaoyu Peng, Xingpeng Wang, Yaping Dong, Wu Li brings up a new data about influence of sodium carbonate and sodium hydroxide on solubility, metastable zone on borax nucleation, however some findings need to be clarified in more details. In my opinion it can be accepted for publication, but after the following revisions:

- 1) Authors should have explained why did they use such a high cooling rates and why is analyzed only a narrow pH range (from 10.1 to 10.5)?
- 2) The English style should be revised; some grammatical errors are found. It is especially wrong that authors use word absorption instead of adsorption (also strange phrases were used through the whole manuscript for example ..heated with the settled heating rate - heated with the certain or adjusted heating rate...
- 3) On the page 4 rows 37-40 it is stated that solubility increases with mass fraction of sodium carbonate which is not in accordance with results presented in Fig 3. In the next sentence they claim that solubility decreases as is shown in figure...
In order to have better comparison of the results I would advise the authors to set the vertical axis in figure 3a and 3b to the same scale.
- 4) Page 5 rows 27-30 I think authors need check values of enthalpy and entropy obtained from literature, because the values cited in the text and given in the table 3 denoted with asterisk differs.
- 5) Page 5 rows 37-38 ...higher values of ΔH and ΔS , which is consistent with general thermodynamic principles... I suggest authors to provide, cite this principle.
- 6) page 7 rows 48-50 The authors said that the effect of increment of the mass fraction of sodium carbonate on borax solubility were uniform, I think this is not an appropriate explanation and that it would be better to say that increment in mass fraction of sodium carbonate causes a gradually decrement of borax solubility.
- 7) As far as I understand (sentence is not clear enough (page. 7, lines 42-45) Authors assumed that sodium carbonate, added in the mother liquor, absorbs on the surface of the nucleus and then suppress activities of nuclei in solution, - It should be explained what nuclei activities are supposed to be; is that nuclei growth? Do they think that sodium carbonate will adsorb on the nucleus surface or will only one of the ions of this molecule be adsorbed? It is difficult to accept the assumption that the ions, added to the mother liquor, were absorbed on the nucleus surface and suppressed nucleation, while the RDX pattern of the grown crystal did not show a difference in structure compared to the crystals grown without added ions. I think the authors need to discuss this fact in more detail.

Sincerely,

Review form: Reviewer 2

Is the manuscript scientifically sound in its present form?

Yes

Are the interpretations and conclusions justified by the results?

Yes

Is the language acceptable?

Yes

Is it clear how to access all supporting data?

Yes

Do you have any ethical concerns with this paper?

No

Have you any concerns about statistical analyses in this paper?

No

Recommendation?

Accept with minor revision (please list in comments)

Comments to the Author(s)

This study seems to be carefully carried out. However, some minor revisions are necessary:

Abstract; The sentence "It was found the consistent tendency---" is not at all clear.

Page 4: "the solubility of borax increased both..." it seems to me that this statement could be wrong.

Page 4: last paragraph - it should be "according". Next sentence: this cannot be noted from Fig. 3(b), since there is no sodium carbonate!

Page 5: Second paragraph: it should be Table 3.

Next paragraph: it should be "The calculated dissolution..."

Page 6: Third paragraph: it should be "be represented"

Conclusion: it should be "adsorbed on nuclei", not "absorbed"

third line from the bottom: it should be "fraction of sodium carbonate was larger."

Table 4, heading: one time "in different" is enough.

Fig. 2: the formulas are misleading, since they suggest an incorporation of sodium carbonate and sodium hydroxide, which is obviously not the case,

Fig. 3: legend-the numbers in carbonate should be subscripts.

Fig.4. legend-the 0 is also subscript.

Fig. 5 legend: one time R=55 K/h is enough.

Decision letter (RSOS-181862.R0)

19-Mar-2019

Dear Dr Dong:

Title: Effects of CO₃²⁻ and OH⁻ on the Solubility, Metastable Zone Width, and Nucleation Kinetics of Borax Decahydrate

Manuscript ID: RSOS-181862

The editor assigned to your manuscript has now received comments from reviewers. We would like you to revise your paper in accordance with the referee and Subject Editor suggestions which can be found below (not including confidential reports to the Editor). Please note this decision does not guarantee eventual acceptance.

Please submit your revised paper before 11-Apr-2019. Please note that the revision deadline will expire at 00.00am on this date. If we do not hear from you within this time then it will be assumed that the paper has been withdrawn. In exceptional circumstances, extensions may be possible if agreed with the Editorial Office in advance. We do not allow multiple rounds of revision so we urge you to make every effort to fully address all of the comments at this stage. If deemed necessary by the Editors, your manuscript will be sent back to one or more of the original reviewers for assessment. If the original reviewers are not available we may invite new reviewers.

On behalf of the Subject Editor Professor Anthony Stace and the Associate Editor Professor Hazel Cox.

RSC Associate Editor:
Comments to the Author:
(There are no comments.)

RSC Subject Editor:

Comments to the Author:
(There are no comments.)

Reviewers' Comments to Author:
Reviewer: 1

Comments to the Author(s)

The manuscript: Effects of CO₃²⁻ and OH⁻ on the Solubility, Metastable Zone Width, and Nucleation Kinetics of Borax Decahydrate written by Jing Chen, Jiaoyu Peng, Xingpeng Wang, Yaping Dong, Wu Li brings up a new data about influence of sodium carbonate and sodium hydroxide on solubility, metastable zone on borax nucleation, however some findings need to be clarified in more details. In my opinion it can be accepted for publication, but after the following revisions:

- 1) Authors should have explained why did they use such a high cooling rates and why is analyzed only a narrow pH range (from 10.1 to 10.5)?
- 2) The English style should be revised; some grammatical errors are found. It is especially wrong that authors use word absorption instead of adsorption (also strange phrases were used through the whole manuscript for example ..heated with the settled heating rate – heated with the certain or adjusted heating rate...
- 3) On the page 4 rows 37-40 it is stated that solubility increases with mass fraction of sodium carbonate which is not in accordance with results presented in Fig 3. In the next sentence they claim that solubility decreases as is shown in figure...
In order to have better comparison of the results I would advise the authors to set the vertical axis in figure 3a and 3b to the same scale.
- 4) Page 5 rows 27-30 I think authors need check values of enthalpy and entropy obtained from literature, because the values cited in the text and given in the table 3 denoted with asterisk differs.
- 5) Page 5 rows 37-38 ...higher values of ΔH and ΔS , which is consistent with general thermodynamic principles... I suggest authors to provide, cite this principle.
- 6) page 7 rows 48-50 The authors said that the effect of increment of the mass fraction of sodium carbonate on borax solubility were uniform, I think this is not an appropriate explanation and that it would be better to say that increment in mass fraction of sodium carbonate causes a gradually decrement of borax solubility.
- 7) As far as I understand (sentence is not clear enough (page. 7, lines 42-45) Authors assumed that sodium carbonate, added in the mother liquor, absorbs on the surface of the nucleus and then suppress activities of nuclei in solution, - It should be explained what nuclei activities are supposed to be; is that nuclei growth? Do they think that sodium carbonate will adsorb on the nucleus surface or will only one of the ions of this molecule be adsorbed? It is difficult to accept the assumption that the ions, added to the mother liquor, were absorbed on the nucleus surface and suppressed nucleation, while the RDX pattern of the grown crystal did not show a difference in structure compared to the crystals grown without added ions. I think the authors need to discuss this fact in more detail.

Sincerely,

Reviewer: 2

Comments to the Author(s)

This study seems to be carefully carried out. However, some minor revisions are necessary:
Abstract; The sentence "It was found the consistent tendency---" is not at all clear.

Page 4: "the solubility of borax increased both..." it seems to me that this statement could be wrong.

Page 4: last paragraph - it should be "according". Next sentence: this cannot be noted from Fig. 3(b), since there is no sodium carbonate!

Page 5: Second paragraph: it should be Table 3.

Next paragraph: it should be "The calculated dissolution..."

Page 6: Third paragraph: it should be "be represented"

Conclusion: it should be "adsorbed on nuclei", not "absorbed"

third line from the bottom: it should be "fraction of sodium carbonate was larger."

Table 4, heading: one time "in different" is enough.

Fig. 2: the formulas are misleading, since they suggest an incorporation of sodium carbonate and sodium hydroxide, which is obviously not the case,

Fig. 3: legend-the numbers in carbonate should be subscripts.

Fig.4. legend-the 0 is also subscript.

Fig. 5 legend: one time $R=55$ K/h is enough.

Author's Response to Decision Letter for (RSOS-181862.R0)

See Appendix A.

RSOS-181862.R1 (Revision)

Review form: Reviewer 1

Is the manuscript scientifically sound in its present form?

Yes

Are the interpretations and conclusions justified by the results?

Yes

Is the language acceptable?

No

Is it clear how to access all supporting data?

Yes

Do you have any ethical concerns with this paper?

No

Have you any concerns about statistical analyses in this paper?

I do not feel qualified to assess the statistics

Recommendation?

Accept with minor revision (please list in comments)

Comments to the Author(s)

The comments are in the attached file 2nd revision RSOS_2 (Appendix B).

Review form: Reviewer 2

Is the manuscript scientifically sound in its present form?

Yes

Are the interpretations and conclusions justified by the results?

Yes

Is the language acceptable?

Yes

Is it clear how to access all supporting data?

Yes

Do you have any ethical concerns with this paper?

No

Have you any concerns about statistical analyses in this paper?

No

Recommendation?

Accept as is

Comments to the Author(s)

This manuscript has been revised in a satisfactory way!

Decision letter (RSOS-181862.R1)

24-Apr-2019

Dear Dr Dong:

Title: Effects of CO₂- and OH- on the Solubility, Metastable Zone Width, and Nucleation Kinetics of Borax Decahydrate
Manuscript ID: RSOS-181862.R1

Thank you for submitting the above manuscript to Royal Society Open Science. On behalf of the Editors and the Royal Society of Chemistry, I am pleased to inform you that your manuscript will be accepted for publication in Royal Society Open Science subject to minor revision in accordance with the referee suggestions. Please find the reviewers' comments at the end of this email.

The reviewers and handling editors have recommended publication, but also suggest some minor

revisions to your manuscript. Therefore, I invite you to respond to the comments and revise your manuscript.

Because the schedule for publication is very tight, it is a condition of publication that you submit the revised version of your manuscript before 03-May-2019. Please note that the revision deadline will expire at 00.00am on this date. If you do not think you will be able to meet this date please let me know immediately.

Best wishes,

Dr Laura Smith
Publishing Editor, Journals

RSC Associate Editor:
Comments to the Author:
Reviewer 1 has requested some improvements to the language.

RSC Subject Editor:
Comments to the Author:
(There are no comments.)

Reviewer comments to Author:
Reviewer: 2

Comments to the Author(s)
This manuscript has been revised in a satisfactory way!

Reviewer: 1

Comments to the Author(s)
The comments are in the attached file 2nd revision RSOS_

Author's Response to Decision Letter for (RSOS-181862.R1)

See Appendix C.

Decision letter (RSOS-181862.R2)

22-May-2019

Dear Dr Dong:

Title: Effects of CO₃²⁻ and OH⁻ on the Solubility, Metastable Zone Width, and Nucleation Kinetics of Borax Decahydrate
Manuscript ID: RSOS-181862.R2

It is a pleasure to accept your manuscript in its current form for publication in Royal Society Open Science. The chemistry content of Royal Society Open Science is published in collaboration with the Royal Society of Chemistry.

RSC Associate Editor
Comments to the Author:
The revisions are sufficient, and the work can now be accepted.

Reviewer(s)' Comments to Author:

Appendix A

Response to Reviewers' comments

(Manuscript ID: RSOS-181862)

Title: Effects of CO_3^{2-} and OH^- on the Solubility, Metastable Zone Width, and Nucleation Kinetics of Borax Decahydrate

Dear editor:

Thank you for giving us a chance to revise the paper. The comments from the referees are very valuable and instructive to correct and improve our paper. We have suitably addressed their comments or stated our opinions, and then outlined each change point by point. We have learned a lot through the revision process, including the writing, arrangement, and details of the experiment. We appreciate for Editors and Reviewers' warm work earnestly, and hope that the corrections will meet with approval. Revised portion is marked in blue in the revised manuscript. The main corrections in the paper and the responds to the reviewer's comments are as flowing:

Referee: 1

1) Authors should have explained why did they use such a high cooling rates and why is analyzed only a narrow pH range (from 10.1 to10.5)?

Answer:

Thanks to the referee. The cooling rate and heating rate influence the temperature of nucleation and dissolution, respectively. Meanwhile, the MZW decreases when decreasing the cooling/heating rate. Thus, the cooling rate per cycle is the same as the heating rate. By comprehensive consideration of the properties of the chosen system, the stability of the apparatus and the time for experimentation, we have chosen five different cooling/ heating rates (55 K/h, 45 K/h, 35 K/h, 25 K/h and 15 K/h). The main aim of using the different cooling/heating rates is to obtain the saturation temperature by extrapolation to 0 K/min of T_2 - R lines. It was found that the cooling/heating rate we selected is appropriate and consistent with the actual crystallization process.

Based on the changes of CO_3^{2-} in boron containing carbonate-type brine

(Hydrometallurgy,2014,149,143-147), we have chosen the addition of Na_2CO_3 from 0 to 9.22%. In the study we found that the pH changes with the addition of Na_2CO_3 . As pH also have effect on the solubility and MZW of borax, it is suggested to study the effect of pH.

2) The English style should be revised; some grammatical errors are found. It is especially wrong that authors use word absorption instead of adsorption (also strange phrases were used through the whole manuscript for example ..heated with the settled heating rate – heated with the certain or adjusted heating rate...

Answer:

Thanks to the referee. The English style have been checked and corrected.

3) On the page 4 rows 37-40 it is stated that solubility increases with mass fraction of sodium carbonate which is not in accordance with results presented in Fig 3. In the next sentence they claim that solubility decreases as is shown in figure...

In order to have better comparison of the results I would advise the authors to set the vertical axis in figure 3a and 3b to the same scale.

Answer:

Thanks to the referee. The Y-axis of inset graph in figure 3a and 3b were set to the same scale now.

4) Page 5 rows 27-30 I think authors need check values of enthalpy and entropy obtained from literature, because the values cited in the text and given in the table 3 denoted with asterisk differs.

Answer:

Thanks to the referee. We have checked the values and corrected them.

5) Page 5 rows 37-38 ...higher values of ΔH and ΔS , which is consistent with general thermodynamic principles... I suggest authors to provide, cite this principle.

Answer:

Thanks to the referee. We have added a citation.

6) page 7 rows 48-50 The authors said that the effect of increment of the mass

fraction of sodium carbonate on borax solubility were uniform, I think this is not an appropriate explanation and that it would be better to say that increment in mass fraction of sodium carbonate causes a gradually decrement of borax solubility.

Answer:

Thanks to the referee. We have modified as the suggestion.

7) As far as I understand (sentence is not clear enough (page. 7, lines 42-45) Authors assumed that sodium carbonate, added in the mother liquor, adsorbs on the surface of the nucleus and then suppress activities of nuclei in solution, - It should be explained what nuclei activities are supposed to be; is that nuclei growth? Do they think that sodium carbonate will adsorb on the nucleus surface or will only one of the ions of this molecule be adsorbed? It is difficult to accept the assumption that the ions, added to the mother liquor, were adsorbed on the nucleus surface and suppressed nucleation, while the XRD pattern of the grown crystal did not show a difference in structure compared to the crystals grown without added ions. I think the authors need to discuss this fact in more detail.

Answer:

Thanks to the referee. Crystallization is an important separation and purification process in the chemical industry. However, the mechanisms of impurities are not completely clear. Theoretical thoughts indicate the blockade of active growth sites by impurity molecules which adsorb at kink sites of growing crystals. This idea was introduced by Cabrera and Vermilyea (**The Growth of Crystals from Solution, in Perfection of Crystals, Wiley, New York, 1958, 393–407**) and is called the pinning mechanism. Later Kubota and Mullin (**J. Cryst. Growth, 1995, 152, 203–208**) expanded those thoughts by introducing adsorption isotherms like the Langmuir one into the model.

As a critical window, the MZW was employed to explore how sodium carbonate affects the nucleation behavior of borax in our experiment. According to the classical three-dimensional nucleation theory, the value of solid-liquid interfacial energy γ was calculated by MZW data. Our results show that the values of γ increased with increasing Na_2CO_3 content when the saturation temperature and cooling rate were

fixed. Basically, a higher interfacial energy means a bigger nuclear barrier which lead to a harder nucleation process. This proposed a possible explanation as to why Na_2CO_3 could increase the MZW of borax. Besides, the addition of Na_2CO_3 is unsaturated which just had influence on the crystallization process of borax and didn't entered into the structure of borax crystal. Therefore, the XRD pattern of the grown crystal did not show a difference compared to the crystals grown in pure solution.

Referee: 2

This study seems to be carefully carried out. However, some minor revisions are necessary:

Abstract; The sentence "It was found the consistent tendency---" is not at all clear.

Page 4: "the solubility of borax increased both..." it seems to me that this statement could be wrong.

Page 4: last paragraph - it should be "according". Next sentence: this cannot be noted from Fig. 3(b), since there is no sodium carbonate!

Page 5: Second paragraph: it should be Table 3. Next paragraph: it should be "The calculated dissolution..."

Page 6: Third paragraph: it should be "be represented"

Conclusion: it should be "adsorbed on nuclei", not "absorbed"; third line from the bottom: it should be "fraction of sodium carbonate was larger". Table 4, heading: one time "in different" is enough.

Fig. 2: the formulas are misleading, since they suggest an incorporation of sodium carbonate and sodium hydroxide, which is obviously not the case,

Fig. 3: legend-the numbers in carbonate should be subscripts.

Fig.4. legend-the 0 is also subscript.

Fig. 5 legend: one time $R=55 \text{ K/h}$ is enough.

Answer:

Thank you very much for your careful review. We have studied comments carefully and tried our best to revise and improve the manuscript and made changes in the manuscript according to the referees' good comments.

Please feel free to contact us with any questions and we are looking forward to your consideration.

Once more thank you very much for your help.

Sincerely yours,

Prof. Yaping DONG

March 31, 2019

Appendix B

I read the Manuscript ID: RSOS-181862.R1

Title: Effects of CO₃²⁻ and OH⁻ on the Solubility, Metastable Zone Width, and Nucleation Kinetics of Borax Decahydrate

Authors: Chen, Jing; Peng, Jiaoyu; Wang, Xingpeng; Dong, Yaping; Li, Wu

Authors have accepted and made the majority of suggestions given by reviewers, except the one, and that is a grammar (language) correction. The manuscript grammatically needs to be improved further. I have suggested many additional language corrections below. However, I definitely think that authors should seek expert assistance for the correct translation of the manuscript into English. I need to stress out that authors definitely need to improve the Abstract since that is the most read part of the paper.

Comments by reviewer

In Abstract

Mass fraction – mass percentage

Measurements of the solubility and metastable zone width (MZW) for borax decahydrate in sodium carbonate and sodium hydroxide aqueous solutions were obtained.

The temperature ranges from 285 K to 315 K by means of the conventional polythermal method while the onset of nucleation was detected by with the turbidity technique. ...

The solubility of borax gradually decreased, Over with the mass fraction percentages of sodium carbonate in when it ranges from 0 to 9.22 %, whereas the MZW at same conditions clearly broadened. The pH value of the solution varied from 9.9 to 10.5 according to the changes caused by the addition of sodium carbonate. It was found that the solubility was lower and MZW was wider at higher pH values. The nucleation parameters of borax were determined from the metastable zone data and analysed in order to explain the trends-obtained why the MZW become wider. According Applying the classical three-dimensional nucleation theory approach, it was found that the addition of carbonate and hydroxide ions led causes that to the value of solid-liquid interfacial energy, γ , increases. It indicated that CO₃²⁻ and OH⁻ ions adsorbed on the nuclei and suppressed nucleation rate.

Page 7

Rows 10-11

....as functions of temperature and the variation in the presence of other salts in the solution[6]....

Rows 27-28

Peng Jiaoyu [12-14] had investigated the influence of KCl, LiCl and K₂SO₄ on solubility and MZW with laser technique.

Page 8

Row 54

.....(S470 Seven Excellence, Mettler Toledo) **and** with the precision of ± 0.05 .

Page 9.

Rows 28-29

.... Second, the mixture was then.....**Then, the mixture was heated...**

Rows 34-36

...appears, **which was detected by sudden increase in turbidity.**

Row 40-41

and a constant stirring rate of 300 rpm ----and **at constant impeller speed** of 300 rpm

Row 50-52

The metastable zone width of borax **is** represented by the maximum undercooling...

Pg 10

Rows 28-30

...from pure water, **aqueous solutions** of sodium hydroxide and sodium carbonate, **respectively.**

Rows 38-39

...demonstrating the.... – **indicating that** the impurity...

Page 11

Rows 6-10

~~According to the mass concentration of Na₂CO₃ of experimental result,~~ the solubility of borax in different **mass percentage** of sodium carbonate (0.0%-9.22%) **in aqueous** solution was determined.

Rows 20-21

which lead **a** dissolution–precipitate equilibrium **to** move toward the ...

Rows 25-26

...indicating **that...**

Rows 30-36

~~In the study we found that~~ **Since that** the pH changes with the addition of Na₂CO₃ ~~As pH~~ **and that** ~~it also have~~ **has** an effect on the solubility and MZW of borax, it is suggested to study the effect of pH

as well. According to the changes caused by Na_2CO_3 (from 0 to 9.22%), ~~we selected the range of pH {from 9.9 to 10.5}~~ **was selected.**

Rows 36-38

of borax ~~within~~ at Na_2CO_3 **concentration of 9.22%** (pH=10.5) was lower than at pH=10.5 ~~at 10.5~~ adjusted by NaOH.

Rows 38/39

It was concluded that the effect of CO_3^{2-} was stronger than OH^- .

It suggests that CO_3^{2-} **has more pronounce effect on borax solubility increment** than OH^- ions.

Page 12

Row 4

Dissolution enthalpy, **$\Delta_{\text{dis}}H$** , and dissolution entropy, **$\Delta_{\text{dis}}S$** ,....

Row 6-9

... When the solubility of borax in sodium carbonate and sodium hydroxide solution at different temperatures is available, ~~they can~~ than the values of **$\Delta_{\text{dis}}H$** and **$\Delta_{\text{dis}}S$** can be ~~obtained~~ determined from the van't Hoff equation as ...

Row 22-23

~~And~~ Then the dissolution enthalpy and entropy of borax shown in Table 3 ~~can be~~ **were** calculated from the slope and the interception of these plots.

Page 13

Row 23-24

The MZW data of borax **against saturation temperature at** in different **mass percentages** of sodium carbonate was ~~plotted~~ given in Fig. 5 (a).

Row 49

The results of investigation of the influence of pH on MZW of borax ~~was also investigated, as showed~~ are given in Fig. 5 (b).

Row 53

It ~~had~~ **has** long been known **that**

Page 14

Row 7/8

....~~we~~ it can **be** easily find that the borax MZW in Na_2CO_3 -

Row 31-33

Fig. 5. Changes in metastable zone width (R=55 K/h): (a) at different concentration mass percentages of sodium carbonate solutions; (b) at same pH.

Row 54

.....mass percentage fraction

Page 15

Row 6

Where—were F, B (or R) and Z which present _____, _____, _____ value are calculated by equations:

Fig. 6. The plot of $(T_0 / \Delta T_{max})^2$ and vs. $\ln R$ for borax in at different mass percentages concentration of sodium carbonate:

(a) 0.00% Na₂CO₃; (b) 5.31% Na₂CO₃; (c) 7.09% Na₂CO₃; (d) 8.38% Na₂CO₃; (e) 9.22% Na₂CO₃.

Page 16

Row 3-5

Table 4. Values of kinetic parameters of borax at different mass percentages of sodium carbonate estimated using classical three dimensional nucleation theory

Page 17

Row 26/28

Fig.7. The plot of $(T_0 / \Delta T_{max})^2$ vs. $\ln R$ for borax at different pH: (a) pH=10.1; (b) pH=10.3; (c) pH=10.5.

Table 5. Values of kinetic parameters of borax at different pH values estimated using classical three dimensional nucleation theory

Row 58

.....in previous literature previously published paper....

References

Authors should double check the References according to the Vancouver Style. The guidelines are at the website: <https://royalsocietypublishing.org/rsos/for-authors>

For example: [1] Elbeyli İ.Y. 2015 Production of crystalline boric acid and sodium citrate from borax decahydrate. Hydrometallurgy, **158**, 19-26.

Sincerely,

Appendix C

Response to Reviewers' comments

(Manuscript ID: RSOS-181862)

Title: Effects of CO_3^{2-} and OH^- on the Solubility, Metastable Zone Width, and Nucleation Kinetics of Borax Decahydrate

Dear editor:

Thank you for accepting our manuscript. We have studied comments carefully and tried our best to revise and improve the manuscript and made changes in the manuscript according to the referees' good comments. We had rewritten the Abstract to make it better. About the style of the references, we used the Open Biology style file instead. Other revised portions are marked in blue in the revised manuscript.

Thanks for your careful review again. Please feel free to contact us with any questions and we are looking forward to your consideration.

Once more thank you very much for your help.

Sincerely yours,

Prof. Yaping DONG

April 29, 2019